# Creating a culture, not just a space—A qualitative investigation into reflective practice groups in inpatient mental health settings from the perspectives of facilitators and attendees

**Pui Lok Joshua Yiu**[1], **Abbie McDonogh**[1], **Harpreet Gill**[2], **Jo Billings**[1]*

1 Division of Psychiatry, University College London, London, United Kingdom, 2 Camden and Islington NHS Foundation Trust, London, United Kingdom

* j.billings@ucl.ac.uk

## Abstract

### Background

Working in inpatient mental health settings is often characterised by highly emotive work and staff shortages. Despite the suggested benefits of reflective practice groups on staff well-being and clinical practice across healthcare settings, to date, there have been limited empirical studies on reflective practice groups in inpatient mental health settings, especially on group engagement and improvement.

### Methods

We interviewed fifteen participants, including both facilitators and attendees of reflective practice groups. Participants were from eight inpatient mental health wards across two National Health Service settings in the UK. We analysed interview transcripts using thematic analysis.

### Result

We deductively organised the data into themes and subthemes under three overarching domains—"Impact", "Factors on Engagement", and "Improvement". Theme development was generated inductively from the data. For impacts, we found reflective practice groups may bolster staff reflective capacity and team cohesion. The groups may help attendees create appropriate distance from their emotions and overcome power hierarchies. We discovered that the availability of reflective practice, sense of containment in groups, and team composition may influence group facilitation and engagement. For improvements, different measures could be adopted to improve access and engagement of staff with difficulties attending. Facilitators may benefit from more support to establish a reflective culture and experiment with new ways of facilitating.

**Data Availability Statement:** Due to the personal nature of the participant accounts shared in the interviews, and in line with the ethical approval

granted for this study, the full data set of interviews has not been made publicly available. Excerpts of interviews can be made available on reasonable request to the corresponding author via the UCL Research Ethics Committee: ethics@ucl.ac.uk.

**Funding:** The author(s) received no specific funding for this work.

**Competing interests:** No competing interests.

## Discussion

Our findings add to the growing evidence base of the potential value of reflective practice groups in inpatient settings and elaborate on novel mechanisms of their potential impact. This study highlights changeable factors for engagement, concrete recommendations for improvements, and opportunities for further research.

## Introduction

Inpatient psychiatric units aim to provide intensive and timely care to people experiencing mental health crises and presenting with an apparent risk [1]. Admission is entailed when the treatment and care a person requires can only be provided in a mental health inpatient setting and not in the community [2]. In inpatient mental health settings, staff can be exposed to particularly emotionally demanding situations. High rates of aggressive behaviours directed to staff have been reported [3], associated with factors including severe psychiatric conditions and prolonged or involuntary admission [4]. Staff are also often subject to the distressing experiences of patients, including traumas and self-injurious behaviours [5]. To ensure safety in the ward, staff may have to perform physical restraint and invasive treatment [6]. Inpatient staff also often report staff shortages and excessive administrative duties which can create significant stress [7].

Considering the highly stressful working environment in inpatient psychiatric wards, the British Psychological Society (BPS) and the Association of Clinical Psychologists (ACP) have recommended the use of reflective practice groups to promote the clinical understanding and well-being of inpatient staff [8, 9]. Reflective practice involves critically analysing daily working practices and examining reasons behind current and alternative ways of practice, thereby enhancing professional competence [10–12]. Reflective practice groups in healthcare settings are typically group discussions, often facilitated by clinical psychologists or psychotherapists, to encourage reflection on clinical practice. Despite nuances in definitions of reflective practice groups [13–15], the groups often focus on reflection on relational aspects in clinical practice [16]. Kurtz proposed running groups in teams in healthcare settings regularly for at least forty-five minutes [17].

More empirical evidence, however, is required to underpin the recommendation of running reflective practice groups in inpatient psychiatric settings. To date, most studies on reflective practice groups have been conducted in general healthcare settings. They have suggested several benefits on clinical practice and staff well-being, including containment of stress and anxiety [18], enhanced empathy, facilitated case formulations [19], improved team relationships [20] and self-efficacy in delivering effective care [21]. However, as these studies were not specific to inpatient psychiatric settings which differ from community or general healthcare settings in their targeted population and level of care, their results may not be directly transferrable. They may at best suggest the potential of reflective practice groups to promote staff well-being and high-quality service provision in inpatient mental health settings.

In comparison, there has been minimal empirical research investigating reflective practice groups in inpatient mental health settings [22, 23], suggesting a limited understanding of their effectiveness and the potential influencing factors in this particular setting. Most literature to date is in the form of descriptive papers that outline principles of facilitation and anticipated impacts of reflective practice groups [24–29], which may not have evaluated the utility and challenges of running the groups in real-life inpatient psychiatric settings.

There are, to date, a few published quantitative studies using self-devised or adapted questionnaires to examine the impacts of reflective practice groups in inpatient mental health settings. One cross-sectional study noted staff-reported benefits of reflective practice groups on their clinical practice and self-awareness [30]. Another single-arm longitudinal study suggested that reflective practice groups might bolster confidence and team cohesion [31]. Despite preliminary quantitative evidence on reflective practice groups' potential effectiveness, these studies could not capture areas of experience not included in the questionnaires, limiting the generation of new understanding in this relatively unexplored field. For example, questionnaires in Green & Cappleman's study explored positive impacts on individuals but not impacts on the team or organisation [30]. Additionally, most questionnaires assessed impacts but not the processes leading to the impacts and factors affecting engagement, which require further investigation [22].

Given their inductive nature, qualitative studies may generate more nuanced and novel understandings of the rich experience of reflective practice groups [32]. To date, there has been very limited qualitative research into this topic [23]. Existing studies have primarily explored the positive impacts of reflective practice groups on areas including emotional well-being and team cohesion [31, 33], but not situations where the groups may not be entirely productive. There has been minimal discussion of factors on engagement [33, 34], with existing research focused on practical barriers like staff shortages without consideration of other factors that may promote engagement [22, 35]. Moreover, to date, no qualitative studies have focused on exploring potential improvements of reflective practice groups in inpatient mental health settings.

Existing qualitative studies are also subject to methodological limitations. No study to date has interviewed both facilitators and attendees and integrated their views into the analysis [31, 33–37], with only one Australian study that interviewed attendees and studied the field notes of facilitators [23]. Additionally, there has been minimal investigation into the experiences of clinical psychologists or psychotherapists who are facilitating reflective practice groups, except in Heneghan's study [22], limiting our understanding of the difficulties they might experience and the support they might require. Most studies only recruited participants from one or two wards [34, 36], reducing the transferability of their results given various cultures and acuities across settings. Notably, some studies only involved nurses but not multi-disciplinary staff or staff at varying career stages [23, 33], limiting understanding from an organisational perspective. Moreover, most existing qualitative studies have obtained very small sample sizes which may not suffice to provide diverse perspectives.

Therefore, this study aimed to explore the experience of reflective practice groups in inpatient mental health settings by combining the perspectives of both facilitators and attendees, with the following research questions:

1. What, if any, impacts do the groups have on attendees and their practice?

2. What factors may affect the attendance of and engagement in the groups?

3. What are some potential opportunities for improvement?

## Methods

### Ethical approval

This study was approved by the Research Ethics Committee of University College London (reference number: 24809/001). Written informed consent was obtained from all participants.

This study was also registered with the Camden and Islington NHS Trust Information Governance Department as a service evaluation. While interviews might raise some emotive issues for staff related to the difficulties of work, the interview guides primarily revolved around staff experiences of reflective practice groups with minimal discussion of potentially distressing information. Participants were reminded of their right to withdraw from the research at any time before commencing the interview and were signposted to sources of support in the Participant Information Sheet if needed.

## Participants and procedures

Participants were mental health professionals working in inpatient services in one central London NHS Mental Health Trust. They were either clinical psychologists with experiences facilitating reflective practice groups or other mental health professionals having attended the groups. Participants were purposively sampled to reflect diversity in ethnic background, career stage, and experience with work and reflective practice groups in inpatient mental health services, capturing a wide range of experience.

Participants were approached through the networks of the researchers. Potential participants were invited to contact the first author, PLJY, via email for more information and to arrange an interview if they were interested in participating. They were sent information about the study and the Participant Information Sheet by return. Potential participants received a maximum of two further email invitations to ensure no undue pressure to participate was incurred. The recruitment started on 10th June 2023 and ended on 31st August 2023.

## Interview guide

We developed two interview guides informed by previous literature for facilitators and attendees respectively. Two principal researchers who were both consultant clinical psychologists with extensive experience working in NHS mental health services provided additional feedback. Both guides covered the potential impacts from reflective practice groups and opportunities for improvement. The facilitator interview guide additionally covered experiences of organising and facilitating the groups, whereas the attendee interview guide captured experiences of accessing and engaging in the groups. Interview guides are included in S1 File.

## Data collection

Interviews were arranged at a time convenient for interested participants, either in person at the inpatient facility or online via MS Teams. Before the interview, we discussed any queries participants might have, and then participants provided written informed consent for research participation. Participants were also asked to choose a pseudo-name to represent their identity in the transcripts. The first author conducted all interviews, which were audio-recorded and transcribed verbatim for subsequent analysis.

## Data analysis

We performed a reflexive thematic analysis [38, 39] on interview transcripts. The analysis was facilitated by NVivo 14, incorporating data from both facilitators and attendees to obtain a holistic understanding of the experience of reflective practice groups. We explored any positive and negative impacts of reflective practice groups, factors affecting attendance and engagement, and opportunities for improvement. Deductive and inductive approaches were employed to generate concepts relevant to the initial research questions and explore unanticipated outcomes and experiences. We engaged in a recursive process of reading, coding,

reflecting, and discussion. We developed common patterns to form preliminary themes inductively to organise data. These themes were gradually elaborated as the iterative process progressed. They were eventually arranged into a hierarchical framework, where deductively obtained over-arching domains connected themes and sub-themes to capture variations. Analysis was first performed by the first author. Other researchers' perspectives were discussed and integrated throughout the cyclic process. We then discussed our findings with participants to examine the credibility of our results and develop a richer reading of our data, further elaborating our results. The themes and subthemes are presented with supporting quotes. Pseudonames chosen by participants are used to protect participants' anonymity.

### Reflexivity

In qualitative research, reflexivity can help demonstrate potential relationships between researchers' personal predispositions and the analysis, allowing examination of the credibility of results [40].

The research team involved in this study was a diverse group representing different career stages, genders, and cultural backgrounds. PLJY and AM are MSc graduates in the field of mental health sciences with no previous experience working in inpatient mental health services. PLJY comes from Hong Kong with experience with another healthcare system. This allowed them to adopt a curious stance with minimal presumptions when collecting and analysing the data. JB is a Consultant Clinical Psychologist and Professor with over twenty years of experience of working in the NHS. HG is a Consultant Clinical Psychologist and the Head of Psychology of Inpatient Services in one central London NHS Mental Health Trust. These supervisors offered insights on organisational and practical considerations that PLJY and AM may not be aware of, especially when examining potential barriers and recommendations. We employed this collaborative and reflexive analysis not to reach a consensus on meaning construction but to develop a more nuanced interpretation of the data.

## Results

### Demographics of participants

A total of fifteen participants were interviewed, including six facilitators who were all clinical psychologists and nine attendees of reflective practice groups from eight inpatient wards across two settings of one London NHS Trust. The wards included psychiatric intensive care units, acute, and rehabilitation wards. The interviewees constitute a diverse group of mental health professionals. Tables 1 and 2 contain participants' demographic information.

### Thematic analysis

We organised the data into three deductive overarching domains–impact, factors affecting attendance and engagement, and improvement. We then inductively developed nine themes and twenty-one subthemes.

**1. Impacts.** This domain describes the positive impacts of reflective practice groups, which may contribute to improved clinical practice and staff well-being (Fig 1).

*1.1 Reflective Capacity*. In inpatient services, clinical supervision sometimes becomes closer to line management and problem-solving instead of an actual reflective supervisory space.

*"Supervision often becomes quite managerial and less about people's feelings about patients."* (Elijah, attendee)

**Table 1. Demographic information of facilitators and attendees.**

| Demographic | | | n |
|---|---|---|---|
| Gender | Male | | 3 |
| | Female | | 12 |
| Age | 25–34 | | 7 |
| | 35–44 | | 4 |
| | 45–54 | | 3 |
| | 55–64 | | 1 |
| Ethnicity | White | British | 7 |
| | | Other | 3 |
| | Black | African | 3 |
| | | Other | 1 |
| | Other | Turkish | 1 |
| Job title | Nursing Staff | Ward Manager | 2 |
| | | Charge Nurse | 2 |
| | | Staff Nurse | 1 |
| | | Assistant Practitioner | 1 |
| | Psychology Staff | Lead Clinical Psychologist | 2 |
| | | Highly Specialist Clinical Psychologist | 1 |
| | | Clinical Psychologist | 3 |
| | | Clinical Associate in Psychology | 3 |

In contrast, interviewees believed the primary aim of reflective practice groups is to promote active emotional and relational reflection. Despite a culture of immediacy in the ward, facilitators often try to contain the urge for quick fixes, from both attendees and facilitators themselves, encouraging participants to remain in active reflection. This is what sets reflective practice groups apart from other spaces.

**Table 2. Experience with reflective practice group in inpatient mental health services.**

| Experience | | n |
|---|---|---|
| Year(s) of working in inpatient mental health services | < 1 | 2 |
| | 1–5 | 7 |
| | 6–10 | 5 |
| | > 10 | 1 |
| Number of facilitated/attended reflective practice groups in inpatient mental health services | 1–5 | 1 |
| | 6–10 | 3 |
| | 11–20 | 1 |
| | > 20 | 10 |
| Frequency of facilitating/attending reflective practice groups in inpatient mental health services | More than once a week | 2 |
| | Once a week | 4 |
| | Once every other week | 4 |
| | Once a month | 4 |
| | Once every other month | 1 |

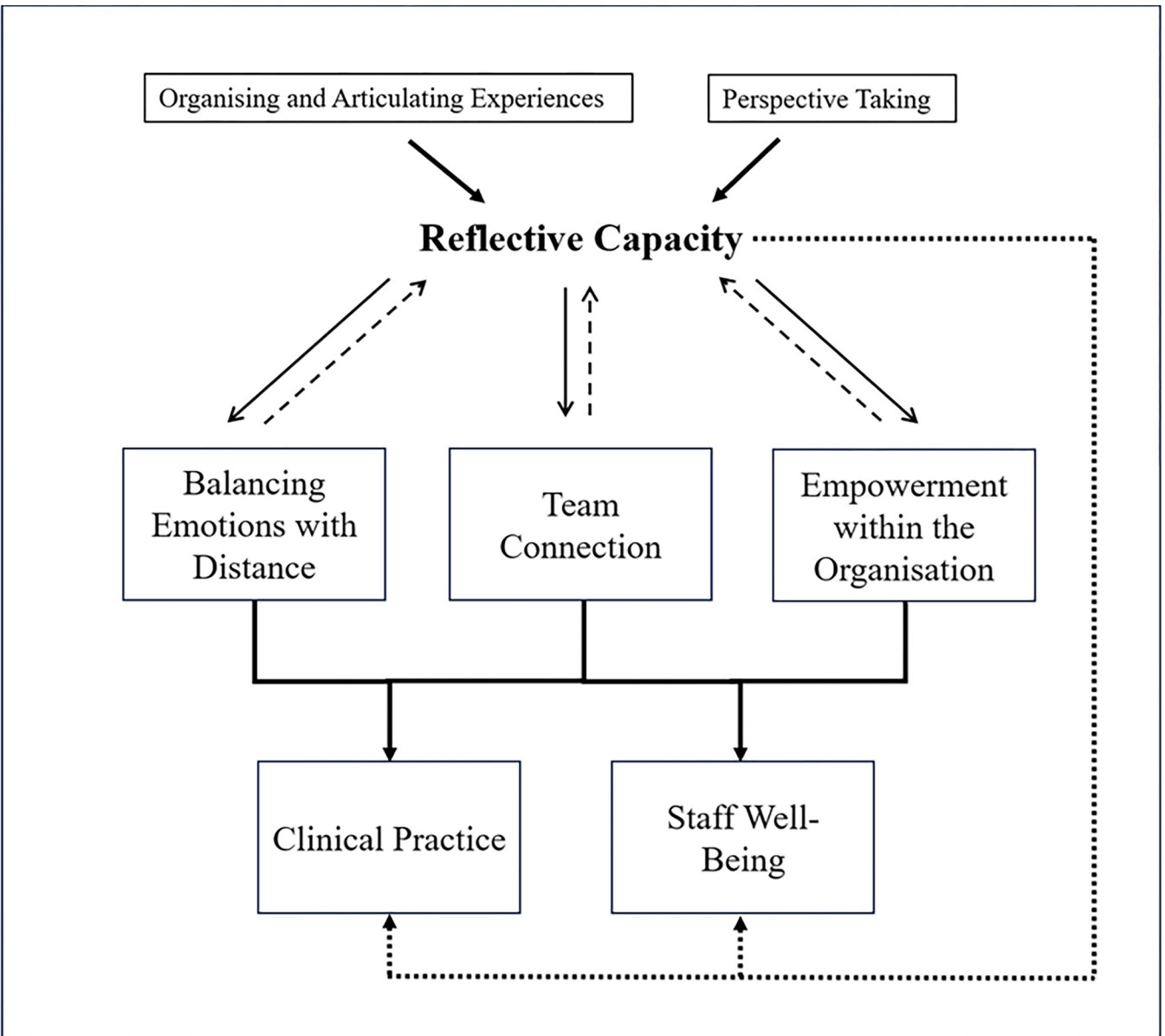

**Fig 1. Impacts of reflective practice group.** Organising and articulating one's experience of work and perspective taking can enhance reflective capacity, which may in turn improve clinical practice and staff well-being through the other three impacts. Solid lines indicate relationships supported by our data. The dotted line linking reflective capacity with clinical practice and staff well-being represents alternative mechanisms not explicit in our data but hypothesised as potential relationships from our knowledge of the literature. The other three dotted lines represent our hypothesised bidirectional relationships between reflective capacity and the three impacts.

*"It's not a business meeting. It's not where we are going to make a lot of decisions. . . I want them to be thinking actively and responding."* (Amelia, facilitator)

During reflective practice groups, reflective capacity may be bolstered through organising and articulating *"a summary of what's happening and how they would think about it and how they would talk about it"* (Charlotte, facilitator). It is also helpful to invite curiosity and perspective-taking, eliciting slow contemplation over some automatic practice without assuming a right or wrong answer:

*"It is very refreshing having maybe the psychologist or new student nurses or new starts being like, 'Oh, why do you do that?'"* (Florence, attendee)

*1.2 Balancing Emotion with Distance.* In a highly emotionally loaded environment, reflective practice groups may enable staff to utilise emotions to benefit their well-being and clinical practice, through the following mechanisms.

*Sub-theme 1*: *Communicating*: Interviewees believed having negative feelings could traditionally be perceived as signs of inadequacy and vulnerability. Reflecting on painful feelings in a group setting may help staff to validate them and model healthy communication at work, without being afraid that the emotional experience is abnormal or problematic.

*"So I try to use self-examples to demonstrate how we can talk about this in a professional way without feeling that you can't say anything bad about each other or the patients."* (Scarlett, facilitator)

*"I'd be like, "Thank God, someone else also feels like this."* (Violet, attendee)

*Sub-theme 2*: *Contextualising*: When attendees start to be in touch with their negative feelings, facilitators may utilise their professional curiosity to help staff to appreciate the intrapersonal, interpersonal, and organisational factors that constitute their emotional experience. This may facilitate their examination of the impacts of emotions and different ways of responding.

*"Talking in a context where you got often a psychologist as a facilitator was very useful because they could help you contextualise what you are experiencing."* (Elijah, attendee)

*Sub-theme 3*: *Containing*: While reflective practice groups can allow ventilation of emotions, interviewees acknowledged that sometimes people might feel stuck or powerless, especially when the concerns involve wider organisational considerations like funding and staffing that might not be changed easily. Facilitators can draw upon different psychological approaches to contain difficult emotions and redirect attendees to the here and now.

*"I think it is a bit of perhaps dialectic. . . sometimes there are also things that we really cannot do much about. And there may still be a way to think about those things that we have control over."* (Mila, facilitator)

*1.3. Team Connection. Sub-theme 1*: *Therapeutic approach of the team*: Working in a fast-paced ward with staff shortages, team clinical practice may at times become automatic and rigid. Here, the groups may help the team to re-appraise patients' responses by integrating historical information, and direct and indirect observation of different members into a more consistent, balanced, and empathetic understanding.

*"They can think more widely and deeply about their service users and their service users' situations, their histories, what has led to the mental health difficulties. . . to bring more understanding and empathy to their work with a service user"* (Amelia, facilitator)

The groups may also help the team to formulate consistent approaches that create a contingency encouraging adaptive behaviours of patients and avoid the misallocation of excessive responsibilities or the development of unhealthy attachments. For example, the team decided how all members should respond to a patient who always asked a specific staff member for support.

*"'I'm the person that's with you at the moment so I'm going to help you' and just be a bit more clear and assertive."* (Scarlett, facilitator)

*Sub-theme 2*: *Improving team morale*: Interviewees found celebrating good practices in reflective practice groups conducive to team morale. Additionally, the groups may promote in-depth understanding and connection between colleagues which might not be easily achieved when working in a fast-paced ward.

*"So you remember like, "Okay, let me check in on that person today because they mentioned this in reflective practice."* (Ava, attendee)

Some interviewees explained how open communication regarding team dynamics may facilitate perspective-taking, enhancing acceptance and respect for diversity.

*"You just get to know how that person will feel about a situation that is different from yours. Like it's okay to feel both ways."* (Honey, attendee)

*1.4. Empowerment within the Organisation. Sub-theme 1*: *Engaging management support*: Attendees noticed that some concerns raised in reflective practice groups were escalated to more senior management, leading to tangible changes that reduce staff pressure.

*"Now they've decided this particular patient cannot be admitted on our ward for some period of time to give a break to the staff team"* (Sofia, attendee)

Despite there often being no immediate changes, some attendees still found their concerns being formally addressed by management validating. Ward managers may also take this opportunity to understand the well-being of individual staff, providing additional support if needed.

*"You don't feel like nobody cares about you. They provide you some genuine reasons why things have not been done to resolve our issues."* (Michael, attendee)

*Sub-theme 2*: *Transcending power dynamics*: Some facilitators described inpatient services as highly hierarchical. Reflective practice groups may empower staff in lower-status roles by equally valuing everyone's perspective and knowledge.

*"The people that are paid the least actually are usually the people that spend the most time with the patients and therefore have the best knowledge of the patients. . . I think consultants and managers learn a lot from being present and hearing the views. . ."* (Scarlett, facilitator)

In the groups, discussing the feelings and reactions of each person may highlight how different team members, regardless of their seniority, banding, and profession, may be impacted by their work similarly, highlighting human commonality and thereby transcending the power hierarchy.

*"I think it's a way of equalising, at least for an hour, the power dynamics and feel like on the same side. . .just seeing that human side of the work. . ."* (Ariel, facilitator)

**2. Factors on attendance and engagement.**   This domain describes systemic factors that may affect group attendance and engagement (Fig 2).

*2.1 Availability of Reflective Practice. Sub-theme 1*: *Facilitator availability*: Interviewees often emphasised how the consistency of reflective practice groups may help set up an expectation of the groups, which may facilitate prior time management and staff allocation, allowing most staff to attend the groups.

*"So that then helps to have that consistency and the nurses on the ward can then complete the allocation appropriately."* (Florence, attendee)

As staff may have different work allocations and unexpected incidents could occur, having more frequent and flexible reflective practice groups could enhance access to the groups.

*"She does it once a week whereas a lot of psychologists will only do once every two weeks. She will make it very available."* (Elijah, attendee)

*Sub-theme 2*: *Support from managers*: Interviewees frequently emphasised the importance of managerial involvement in sorting out practical issues to ensure sufficient staff cover and a private space, allowing staff to attend and concentrate in the groups for an undisturbed period.

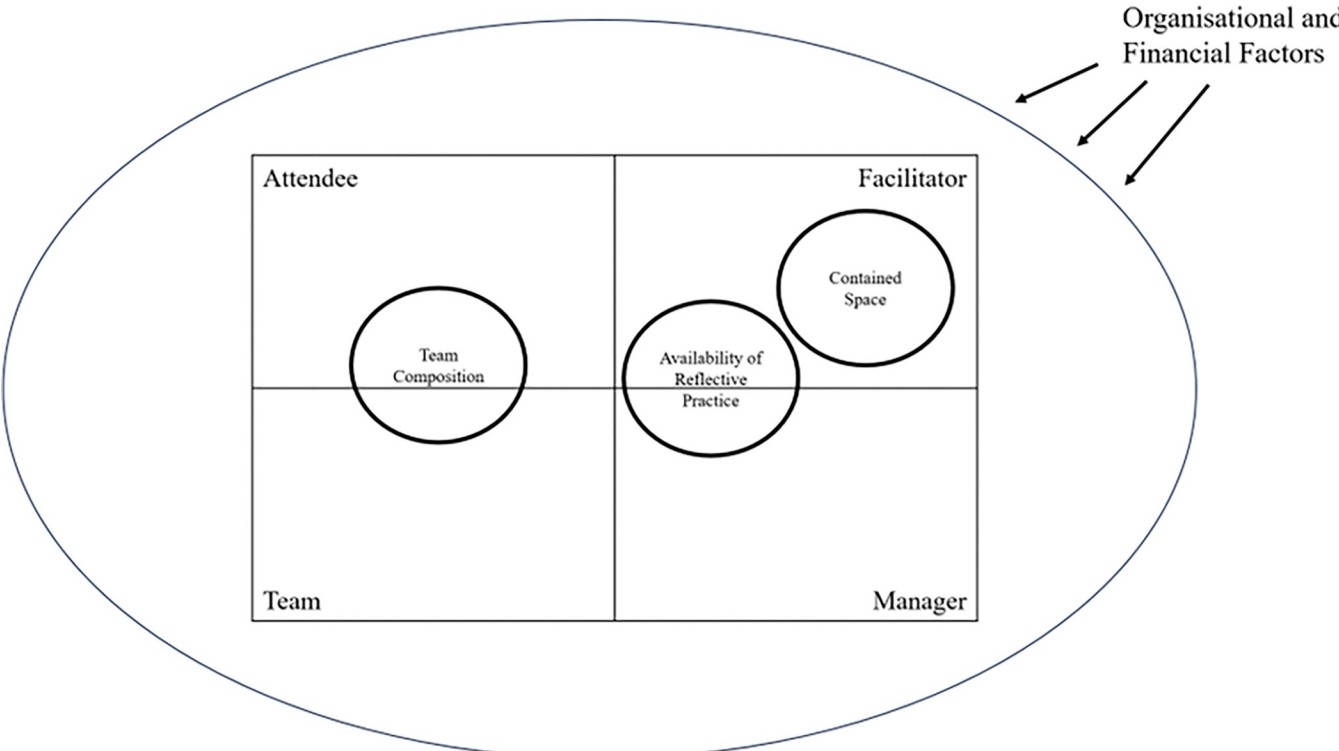

**Fig 2. Factors on group engagement within the organisational and financial context.** For team composition, individual perception of attendees and the presence of a stable group of reflective senior staff may affect engagement. The frequency, flexibility and consistency of facilitators constitute the availability of reflective practice that can affect engagement. Manager's involvement in sorting out practical issues can also enhance the availability. As for the containment in the space, facilitators can promote engagement by fostering a sense of safety and ensuring active participation by different people.

*". . .the ward manager knows everything going during the day so if there is a way to problem solve and try to find out a quick solution, he or she would be the person to do that."* (Emma, facilitator)

*Sub-theme 3*: *Lack of protected time*: Some participants described how, understandably, caring for patients and maintaining patient flow are often prioritised over attending reflective practice groups. Attendees often reported the lack of protected time as the primary difficulty for nursing staff to attend reflective practice groups.

*"So say psychologists, doctors, and occupational therapists, all have protected time for their reflection and their academic learning. This does not exist for nursing staff."* (Elijah, attendee)

Eventually, some felt this boils down to the current allocation of financial resources that limit additional staff cover.

*"It's just financial. . .Somebody just has to pay them that two hours a week, so they have to do it. But we can't do it unless somebody gives us the 'ok'."* (Elijah, attendee)

*2.2 Contained Space. Sub-theme 1*: *Sense of safety*: Attendees often found a strong rapport with facilitators essential in building a sense of safety in reflective practice groups. This rapport might be developed by facilitators' active engagement in the ward, including both formal work and daily interaction.

*"They (staff) know them (facilitators) well, they know how they work and they've built a relationship. So it creates a really safe environment."* (Ava, attendee)

With new starters or senior staff potentially joining each time, interviewees found reiterating ground rules like confidentiality and anonymity useful in creating a sense of safety.

*"There's quite a lot of movement within the team, so it's oftentimes worth repeating some of the ground rules."* (Mila, facilitator)

*Sub-theme 2*: *Active participation by everyone*: Some interviewees underlined the importance of an equal conversation, especially when some people might dominate the conversation. Here, facilitators ought to offer participation to everyone equally.

*"Keep an eye on who is speaking and just offer out open questions to the others or get them to reflect on what someone has said."* (Scarlett, facilitator)

Similarly, some facilitators emphasised the need, potentially with their prior understanding of the team, to respond to any power imbalance or team dynamics that inhibit equal participation. Their roles in addressing power imbalance might sometimes extend outside the group and involve providing debriefs to both parties.

*"if you notice. . . they were using the hierarchy or they were picking up on what staff were saying and kind of turning it into something negative. . . you need to manage that in the conversation and then you need to discuss it with those people afterwards"* (Scarlett, facilitator)

In inpatient wards, facilitators tend to have a more senior position and regularly communicate with ward managers. Facilitators should be mindful of whether they might have

constituted a perceived or an actual power imbalance. Having occasional external facilitators who are perceived as more neutral and independent might also be helpful.

*"At times people have this perception that the resident psychologist might be more inclined to senior managers than ensuring the concerns are resolved."* (Michael, attendee)

*2.3 Team Composition*. *Sub-theme 1*: *Individual perceptions of reflective practice*: Some interviewees described how previous distressful and unfruitful experiences of reflective practice groups may hinder staff engagement. Additionally, limited prior understanding might also prevent staff from meaningfully engaging in the groups. Individual's self-awareness and recent life events may also influence their readiness to be in touch with their internal feelings.

*". . .it's been a place that has felt uncontained, where they may have been confronted by very distressing, difficult experiences between each other."* (Amelia, facilitator)

*"I think it does take a lot of time to understand what it is, what do you expect from it just by observing the routine."* (Michael, attendee)

*Sub-theme 2*: *Senior Staff*: While some believed the presence of senior staff might reduce the sense of safety, other attendees explained how a group of stable and reflective senior staff may help shape a team awareness of reflection and well-being, modelling engagement in the groups themselves.

*"Not putting pressure on people to attend but I guess encouraging it or modelling it by attending themselves."* (Grace, attendee)

Without such senior staff, some facilitators expressed uncertainty about how much to push for attendance and worry about making people feel imposed upon. Here, a strong rapport and tactfulness may enable facilitators to actively overcome ambivalence.

*"I got along with that staff member very well. In a, kind of, humorous way I was like, 'I feel like I must smell or you're avoiding me because, every time I try to do this, you're telling me no.'. . .He was like, 'No Scarlett, that's not my intention. Okay, let's do it at this time. . .'"* (Scarlett, facilitator)

**3. Improvement.**   This domain captures opportunities for improving reflective practice groups. See Table 3.

*3.1 Improving Access and Engagement*. *Sub-theme 1*: *Involving all staff*: Reflective practice groups are often held in the daytime on a regular weekday, which might limit or even exclude some staff from attending the groups. Some interviewees underlined the need for improving access for these staff groups, proposing allowing attendance via Microsoft Teams.

**Table 3. Inductive themes and subthemes under the domain of "Improvement".**

| Improving Access and Engagement | Involving all staff |
|---|---|
| | Documentation |
| Reflection on Reflection | Creating a culture, not just a space |
| | Support for facilitator |

*"But there are some staff that will only do nights and that's for personal or family reasons. . . So then they don't."* (Florence, attendee)

*"And if I don't work on Fridays? Then it means I will be missing out all experience."* (Michael, attendee)

There has been limited participation by professionals like doctors and occupational therapists, who have their own protected time and supervision. Still, interviewees acknowledged the value of the involvement of a whole multidisciplinary team.

*"But I think there's something gained by having everybody thinking about things together, so you can form a whole team view of something happening at the moment."* (Elijah, attendee)

As the groups may involve in-depth emotional and relational reflection and require a certain level of self-awareness, new starters may benefit from additional support in understanding the processes and benefits of the groups or even preparing in advance for the groups.

*"If I am to have a student and who's going to go, I would rather prepare them what is entailed in reflective practice. . . just buy them up right from the onset and they can engage better with it."* (Michael, attendee)

*Sub-theme 2*: *Documentation*: Anonymous documentation may facilitate access to discussed reflection and promote team understanding, especially for staff who could not attend the groups. Revisiting the documentation may also inspire observation and more proactive change in clinical practice.

*"Sometimes we are not all in the same reflective practice and if we can have a written template on what has been discussed, what has been raised, would be good."* (Honey, attendee).

*"Okay, we're seeing consistent themes here. . . what could we be doing as a ward base to be implementing that change because this is where it's happening?"* (Florence, attendee)

*3.2 Reflection on Reflection. Sub-theme 1*: *Creating a culture, not just a space*: The immediacy culture and the blame culture in the ward might limit the safe space, be it internal or external, for staff to engage in slower reflection and express their feelings and thoughts. This effect might be aggravated by a low staff-to-patient ratio or ongoing allegations and incidents, making people *"feel worried about being perceived or judged in a negative way"*. (Arial, facilitator)

Here, a genuine and open acceptance of sub-optimal care and emphasis on continuous improvement may be beneficial, especially when reiterated or even modelled by senior staff or ward managers. This may involve a paradigm shift from assigning responsibility for an incident to learning as a team.

*"Let's not try and hide anything or pretend that everything is perfect and going well. There's no shame in saying, 'Yeah, this happened. Okay, what can we do better next time?'"* (Florence, attendee)

In line with this thinking, the groups might benefit from a shift from solely responsive reflection on incidents for stabilising the situation and preventing repetition, to more proactive reflection on how to help patients progress and replicate previous successes.

*"'How can we help this person leave this ward?' Rather than, 'Today we want to talk about someone who has attacked a lot of people.'"* (Amelia, facilitator)

By redressing these cultures that stifle reflection, some interviewees anticipated growing reflection across the organisation outside reflective practice groups, from *"informal conversations"* (Charlotte, facilitator) to *"end-of-shift reflection"* or even *"nursing supervision"* (Elijah, attendee).

*Sub-theme 2*: *Support for facilitators*: Some facilitators described their experiences of isolation and self-doubt, hinting at a training gap in group facilitation in a highly emotive environment working with patients with severe mental health difficulties. "*Peer supervision*" and "*reflective practice training*" from more experienced facilitators might help bridge the gap.

*"You can feel a bit of imposter syndrome or like, 'Am I doing it right? How are other people doing it? Should I be doing it differently?'"* (Ariel, facilitator)

It would also be beneficial to encourage experimenting with different ways of facilitating and collect more systematic and formal feedback, which facilitators currently may not have other than informal conversations.

*"One possible change could be that you do get an external facilitator to that ward, whether or not that would improve it I'm not sure, but I guess it's something that could be trialed."* (Scarlett, facilitator)

Ariel, a facilitator also talked about how exploring different ways of facilitating may reduce the sense of "*there being a right or wrong way*" and "*needing it be perfect or textbook*". This may align with the core value of reflective practice.

## Discussion

This study aimed to explore attendees' and facilitators' experiences of reflective practice groups in inpatient mental health settings, specifically regarding their perceived impacts, factors on attendance and engagement, and opportunities for improvement. The findings highlight generally positive impacts on reflective capacity, emotional understanding, team cohesion, and redressing hierarchies. The findings also show that the availability of and containment in reflective practice groups and the team's inherent reflective culture may influence attendance and engagement, hinting at areas for improvement.

This study adds to the growing evidence base supporting reflective practice groups in inpatient mental health care by illustrating their potential benefits in improving staff well-being and patient care. This lends further support to the recommendations of the British Psychological Society and Association of Clinical Psychologists to utilise reflective practice groups to support staff well-being regarding their work with chronically distressed service users [8, 9]. In contrast to previous studies which viewed reflective practice groups as clinical group supervision [13, 14], this study highlights the distinctive function of reflective practice groups in promoting critical deliberation of clinical experiences and may help reduce ambiguity in defining reflective practice groups [15]. This study converges with previous studies on the positive impacts of reflective practice groups, namely emotional understanding [22], consistency of team approach [37], team connection [20], and case formulation [19].

By incorporating diverse perspectives from both clinical psychologists and attendees of different career stages, this study provides a more nuanced understanding of impacts when compared to existing literature. For example, Heneghan et al.'s study found that reflective practice

groups can enhance emotional understanding by containing difficult emotions [22]. Our study reveals how reflective practice groups may also validate, contextualise, and model communication of difficult emotions at work, enhancing emotional competence. Additionally, our research shows these groups may boost team morale by celebrating good practice, fostering a sense of connectedness, and improving team dynamics, which have not been explored in previous studies. Moreover, this study discovers novel impacts: empowerment within the organisation by gaining management support, transcending power dynamics, and emphasising common human experiences. These new findings contribute to the limited understanding of the process of reflective practice groups in previous literature [22] and may inform best practice of reflective practice groups.

This study places a strong emphasis on understanding factors affecting engagement, a relatively unexplored area in previous literature. Extending from previous findings regarding the negative impact of the immediacy culture of wards [22] and lack of protected time and space [23] on staff attendance, this study further demonstrates how these factors can also influence attendees' sense of safety and vigilance on the ward, hindering full engagement in reflective practice despite physical attendance. This study also suggests reflective practice groups might mirror cultures in the wider organisation, such as cultures of blame and immediacy. Furthermore, we elaborated on factors not discussed in previous literature, including facilitator availability and integration within the team, containment of power imbalances, individual perceptions, and modelling by senior staff, illuminating opportunities for improvement.

## Implications

The findings have several implications for clinical practice. Concerning staff support, reflective practice could be better explained to new starters during onboarding, which is in line with the NHS onboarding guidance endorsing the enhancement of a sense of safety at work and access to ongoing learning opportunities [41]. Reflective mentoring may facilitate this process given the potential benefits of mentoring in enhancing competence of unfamiliar job content for new nurses [42]. Facilitators and ward managers may actively and tactfully encourage staff with observed reluctance to attend. While participation through online platforms may increase access, confidentiality, sense of safety, and potential distraction should be considered to ensure meaningful engagement. Limited involvement by medical staff could be further explored and addressed to encourage multi-disciplinary participation in reflective practice groups.

Considering the importance of senior members like ward managers and charge nurses in modeling participation in reflective practice groups, the ability to engage in self and group reflection should be encouraged and supported during staff development. Managers may benefit from a clear contract with the facilitator on the expectation from reflective practice groups [17, 25] and should consistently convey the permission for staff to participate. Accessibility to reflective practice groups should be considered in staff allocation especially when arranging shifts. Given the possibility of nursing supervision turning into line management, additional supervision of supervision could be considered to bolster reflection in supervision.

While reflection is an essential clinical skill for all healthcare professionals, the duty of facilitation of reflective practice groups often falls on clinical psychologists and psychotherapists who may be well-placed to provide it. Yet, given an inherent difficulty and a potential training gap in facilitation [22], facilitators may benefit from further training, peer supervision, and their own reflective practice. While co-facilitation by two facilitators might place a greater resource demand on psychologists, this may allow continuity of the groups in the absence of one facilitator and new facilitators to be trained up and gain experience alongside more experienced facilitators. This might also enhance reflection and a sense of containment especially in

a large group or when highly emotive incidents are being discussed. To increase the availability of reflective space, clinical associate psychologists (CAPs) who received training in working with staff teams and leadership skills [43] could be supported in providing brief reflective spaces, such as end-of-shift reflection, which aligns with the role's purpose to serve as a psychological resource for staff [44]. Psychologists may benefit from more systematic and formal feedback from attendees, which could be facilitated by staff with a more neutral role in reflective practice groups, such as CAPs.

One future research direction is to adopt a health economic approach. One major impediment to reflective practice groups is the lack of financial resources, leading to staff shortages [45] and a lack of protected time for continuous professional development. Sufficient cover by experienced staff remains one main determinant in attendance and engagement in a reflective practice group [30]. Notably, there has been poor retention and high turnover for nursing staff [46], with significant economic and non-economic costs [47, 48]. Nurse training, including in-service training and continuous professional development, and mental well-being have been found to significantly impact staff retention in mental health care [49]. With increasing evidence for the benefits of reflective practice groups on staff well-being and clinical practice, future research may adopt a health economic approach and explore the potential of reflective practice groups in reducing costs related to undermined staff well-being, suboptimal patient care, and high turnover. This may inform the allocation of funding to protect time for reflective practice groups.

Furthermore, we also observed some seemingly opposite views in our data, namely regarding the impact of the involvement of an external facilitator and the participation of managerial staff. These findings might not be contradictory but instead underline the complex interplay between different factors in determining the effectiveness of reflective practice groups. Future research may attempt to construct a multifactorial model that informs best practices under different conditions. Future studies may find relevance in attachment and group process theories [22]. For example, given the inherent complex power dynamics in inpatient communities [50], a phase-based approach might account for how the sense of safety and group norm may need to first be developed and consolidated before senior management could engage in the groups meaningfully without stifling discussion, and how qualities of the facilitator and senior staff may influence this process. Such a model may be particularly relevant to settings considering setting up reflective practice groups or facilitators new to their role.

## Strengths and limitations

To our knowledge, this study has to date recruited the most diverse number of participants of reflective practice groups in inpatient mental health settings. This is also the first study in the UK that incorporates the views of both facilitators and attendees through semi-structured interviews. This study also interviewed clinical psychologists whose views are often underrepresented in previous qualitative research. This study also recruited interviewees from different career stages, from assistant practitioner to ward manager for attendees, and from clinical psychologist to lead clinical psychologist for facilitators. The interviewees came from wards with different acuity and populations. While we acknowledge it is arguably impossible to reach an extent of theoretical 'knowingness' where we can confidently assert that no further information will be solicited [38], we deemed this sample as suitable not just because of a larger sample size than the ones in studies to date, which are between three [35] to ten [34], but predominantly because of the population diversity, hence the likely perspectival diversity in the data [51]. This was demonstrated by the generation of previously unexplored themes.

This study moved from focusing on the impacts of reflective practice groups to exploring factors affecting attendance and engagement and potential improvements. Interviewing

participants from different career stages and wards contributed to a complex understanding of challenges faced by professionals from different positions and settings. This allows us to suggest concrete areas for improvement and future research.

Nonetheless, this study is not without limitations. Firstly, this study relied on purposive and snowball sampling, potentially excluding people with less favorable views. This might be one explanation as to why we did not observe potential negative impacts of reflective practice groups. However, some themes concerning group engagement, particularly the ones that touch on an individual's previous experience of uncontained groups and the role of facilitators in containing power imbalance in a hierarchical environment may shed light on how reflective practice groups, when not properly implemented, might lead to potentially adverse impacts. Future studies could try to investigate views from people with previous unfavorable experiences of reflective practice groups and explore potential remedies.

Also, this study did not recruit psychiatrists and occupational therapists, hence the unknown significance of these missing groups. Future studies could try to recruit these professionals and delve into how the composition of participants of different professional backgrounds and seniority may interact with other factors to influence group processes, which might contribute to a potential multifactorial model on group outcomes as previously discussed.

Some might argue that our study did not involve a large number of participants in each subgroup (i.e., facilitators and attendees). However, we aimed to acquire a holistic understanding of reflective practice groups in inpatient psychiatric settings, and we believed the combined perspectives from a reasonably sized sample has achieved this aim. It would be interesting for future studies to recruit a larger number of participants in each subgroup and compare and contrast their views.

## Conclusion

This study adds to the emerging evidence base of the potential value of reflective practice groups in inpatient mental health settings and reveals opportunities to improve engagement. This study also shows how reflective practice groups may reflect the cultures of inpatient services and organisational dynamics in the wider context. Future studies could look into the potential health-economic benefits of reflective practice groups to encourage more funding in securing time and space for reflective practice, potentially enhancing staff well-being and care for service users. Studies could also attempt to develop a multifactorial model to inform best practices of reflective practice groups.

## Supporting information

**S1 File. Interview guides.**
(DOCX)

## Acknowledgments

With thanks to participants who contributed to the data and discussion of findings.

## Author Contributions

**Conceptualization:** Pui Lok Joshua Yiu, Harpreet Gill, Jo Billings.

**Data curation:** Pui Lok Joshua Yiu.

**Formal analysis:** Pui Lok Joshua Yiu, Abbie McDonogh, Jo Billings.

**Investigation:** Pui Lok Joshua Yiu.

**Methodology:** Pui Lok Joshua Yiu, Abbie McDonogh, Jo Billings.

**Project administration:** Pui Lok Joshua Yiu, Harpreet Gill, Jo Billings.

**Resources:** Harpreet Gill, Jo Billings.

**Software:** Pui Lok Joshua Yiu.

**Supervision:** Harpreet Gill, Jo Billings.

**Validation:** Pui Lok Joshua Yiu, Abbie McDonogh.

**Visualization:** Pui Lok Joshua Yiu.

**Writing – original draft:** Pui Lok Joshua Yiu.

**Writing – review & editing:** Pui Lok Joshua Yiu, Abbie McDonogh, Harpreet Gill, Jo Billings.

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
