## [Decision Letter · Decision Letter 0]

31 Jul 2024

PONE-D-24-15162Creating a Culture, Not Just a Space – a Qualitative Investigation into Reflective Practice Groups in Inpatient Mental Health Settings from the Perspectives of Facilitators and AttendeesPLOS ONE

Dear Dr. Yiu,

Thank you for submitting your manuscript to PLOS ONE. After careful consideration, we feel that it has merit but does not fully meet PLOS ONE’s publication criteria as it currently stands. Therefore, we invite you to submit a revised version of the manuscript that addresses the points raised during the review process.

Please submit your revised manuscript by Sep 14 2024 11:59PM. If you will need more time than this to complete your revisions, please reply to this message or contact the journal office at plosone@plos.org. Please include the following items when submitting your revised manuscript:A rebuttal letter that responds to each point raised by the academic editor and reviewer(s). You should upload this letter as a separate file labeled 'Response to Reviewers'.A marked-up copy of your manuscript that highlights changes made to the original version. You should upload this as a separate file labeled 'Revised Manuscript with Track Changes'.An unmarked version of your revised paper without tracked changes. You should upload this as a separate file labeled 'Manuscript'.

We look forward to receiving your revised manuscript.

Kind regards,

Faten Amer, PhD in Health Sciences

Academic Editor

PLOS ONE

Additional Editor Comments:

Dear Dr. Pui Lok Joshua Yiu,

The reviewers have requested revisions for your manuscript.

Please address each point and send the revised manuscript along with a detailed reply to the reviewers' comments point by point by 15 August 2024.

Regards

Reviewers' comments:

Reviewer's Responses to Questions

**Comments to the Author**

1. Is the manuscript technically sound, and do the data support the conclusions?

Reviewer #1: Yes

2. Has the statistical analysis been performed appropriately and rigorously? 

Reviewer #1: N/A

3. Have the authors made all data underlying the findings in their manuscript fully available?

Reviewer #1: No

4. Is the manuscript presented in an intelligible fashion and written in standard English?

Reviewer #1: Yes

5. Review Comments to the Author

Reviewer #1: This was a well written paper that maps onto an identified gap in the literature and makes a useful contribution. Since inpatient mental health settings are often where people report their worst outcomes, interventions that lead to potential better experience and quality improvement are very important to research and evaluate. The comments are therefore focused on improving and strengthening the existing paper.

Minor concerns:

1) Line 123 – the reference number is missing.

2) Line 160 onwards – the section on data analysis needs underpinning, please provide citations linked to the specific approach to thematic analysis and reflection you used here to support transparency.

3) Line 224 – I am assuming these are codenames for the interviewees not actual names but could you please address pseudoanonymisation or anonymisation in the methods explicity?

Major concerns:

1) There is a lack of specificity when writing, that leads to doubt about veracity of the claims being made. This could be removed by tightening up the writing style. For example, the background paragraph opens with “some literature suggested”. What literature? Please use clear citations or try and be more specific and tie the claims made to supporting evidence. I would restructure the entire background section to start with the known problems in the setting, then the claims about the intervention in the setting, before getting to the sentence about what this paper contributes as it will flow better and build the line of argument.

2) When setting the scene and context for reflective practice groups, is there any research showing that these groups are harmful or unhelpful? If you are looking at a particular setting with significant challenges around power imbalances, safety etc. then has this been considered in the literature? Linked to this, in line 405 – are those assumptions or evidence based reasons for hinderance of staff engagement – just be clear when you are writing where the claims are coming from.

3) When reflecting on the number of interview participants, under results – could you please contextualise this i.e. this is average for these types of research or for this topic area/this is a smaller sample because of X or Y reason etc.

4) I think in the discussion and limitations – having nuance around the missing groups or the significance of holding these groups with particular clusters of staff members would be helpful i.e. we do not know if professional background/seniority or what mix of staff facilitates the best outcomes for these groups; we do not know if the mix of staff is as important as the setting and resources to support such groups etc. I think you do have tentative conclusions in there but it would be helpful to indicate more clearly proposed directions for future research.

6. PLOS authors have the option to publish the peer review history of their article (what does this mean?). If published, this will include your full peer review and any attached files.

Reviewer #1: No

---

## [Author Response · Author response to Decision Letter 0]

12 Sep 2024

Dear Editor,

In response to PLOS ONE's decision for the manuscript "Creating a Culture, Not Just a Space – a Qualitative Investigation into Reflective Practice Groups in Inpatient Mental Health Settings from the Perspectives of Facilitators and Attendees" (Decision: Revision required [PONE-D-24-15162] - [EMID:6d0df5fac996c06d]), we would like to thank the editor and reviewer for carefully reading our manuscripts, offering their valuable comments, as well as appreciating the contribution of this manuscript. 

All comments have been thoroughly considered and incorporated into this refined version of our manuscripts, leading to the following changes. We have listed the points raised by the academic editor and reviewer and our response to them. We believe the above changes have addressed the shortcomings in the previous submission. We hope that the new version elicits a positive response.

Best Wishes,

Dr Jo Billings Dr Harpreet Gill

Pui Lok Joshua Yiu Abbie McDonogh 

Editor's comment #1:

Authors' response:

Thank you for providing the templates. The file names, cover page, and titles in the main body have been updated.

Editor's comment #2:

Please include your full ethics statement in the ‘Methods’ section of your manuscript file. In your statement, please include the full name of the IRB or ethics committee who approved or waived your study, as well as whether or not you obtained informed written or verbal consent. If consent was waived for your study, please include this information in your statement as well. 

Authors' response:

Thank you for raising this. I have now included the full name of the ethics committee that approved our study. We have also indicated how we obtained informed written consent under the Methods section (lines 141-143, 159-161, 176-178).

Editor's comment #3:

We note that you have indicated that there are restrictions to data sharing for this study. For studies involving human research participant data or other sensitive data, we encourage authors to share de-identified or anonymized data. However, when data cannot be publicly shared for ethical reasons, we allow authors to make their data sets available upon request. For information on unacceptable data access restrictions, please see http://journals.plos.org/plosone/s/data-availability#loc-unacceptable-data-access-restrictions. 

Authors' response:

Due to the personal nature of the participant accounts shared in the interviews, and in line with the ethical approval granted for this study, the full data set of interviews has not been made publicly available. In response to prompt (a), excerpts of interviews can be made available on reasonable request to the corresponding author via the UCL Research Ethics Committee: ethics@ucl.ac.uk.

Reviewer's comment #1:

Line 123 – the reference number is missing.

Authors' response:

Thank you for pointing this out. This information, alongside the research ethics committee that approved this study, has been added (lines 141-142).

Reviewer's comment #2:

Line 160 onwards – the section on data analysis needs underpinning, please provide citations linked to the specific approach to thematic analysis and reflection you used here to support transparency.

Authors' response:

That's a great reminder. We have now explained the specific approach to thematic analysis we employed, including citation, justification, and our stance on epistemology (lines 183 onwards). 

Reviewer's comment #3:

Line 224 – I am assuming these are codenames for the interviewees not actual names but could you please address pseudoanonymisation or anonymisation in the methods explicitly?

Authors' response:

We appreciate this feedback. We have now explained the process of pseudoanonymisation more explicitly (lines 178-179, 199-200). 

Reviewer's comment #4:

There is a lack of specificity when writing, that leads to doubt about veracity of the claims being made. This could be removed by tightening up the writing style. For example, the background paragraph opens with “some literature suggested”. What literature? Please use clear citations or try and be more specific and tie the claims made to supporting evidence. I would restructure the entire background section to start with the known problems in the setting, then the claims about the intervention in the setting, before getting to the sentence about what this paper contributes as it will flow better and build the line of argument.

Authors' response:

That's a pertinent suggestion. We have now re-examined not just the background section but also other sections to ensure specificity when writing. We have also re-structured the background section following your suggestion - explaining the challenges of working in an inpatient mental health setting, then claims about the utility of reflective practice groups, and what are they, followed by a critique of the limitations of current literature and how our paper could contribute to this literature gap. 

Reviewer's comment #5:

When setting the scene and context for reflective practice groups, is there any research showing that these groups are harmful or unhelpful? If you are looking at a particular setting with significant challenges around power imbalances, safety etc. then has this been considered in the literature? Linked to this, in line 405 – are those assumptions or evidence based reasons for hinderance of staff engagement – just be clear when you are writing where the claims are coming from.

Authors' response:

This is a fresh and useful perspective. Indeed, to date, we are not aware of any literature that has suggested the negative impacts of reflective practice groups. Our data do not directly suggest any either. However, our data do imply settings or conditions (e.g., uncontained power imbalances) under which reflective practice groups may not be productive, or even potentially harmful. In the background section, we have now noted that potential negative impact has not been suggested in any current literature (line 109 onwards). In the discussion section, we reflected on the implications of our data on the potential negative impacts of these groups, as well as how our sampling method might have limited our collection of data that might suggest unhelpful effects of these groups (lines 664-673). 

Reviewer's comment #6:

When reflecting on the number of interview participants, under results – could you please contextualise this i.e. this is average for these types of research or for this topic area/this is a smaller sample because of X or Y reason etc.

Authors' response:

Thank you for this. We have now included the range of sample sizes of existing research in this field and why we consider our sampling was diverse enough to generate new insights, which was linked to our thematic analysis approach (lines 651-657). 

Reviewer's comment #7:

I think in the discussion and limitations – having nuance around the missing groups or the significance of holding these groups with particular clusters of staff members would be helpful i.e. we do not know if professional background/seniority or what mix of staff facilitates the best outcomes for these groups; we do not know if the mix of staff is as important as the setting and resources to support such groups etc. I think you do have tentative conclusions in there but it would be helpful to indicate more clearly proposed directions for future research.

Authors' response:

This is a very helpful direction that sparked some further reflection. The mix of staff is unfortunately an aspect that we were not able to directly explore. This is also likely a controversial one which might imply that certain factors interact and determine whether a mix of staff (e.g., including ward managers or not) is helpful. These factors might include areas we had explored (e.g., the facilitator's ability to contain power imbalance, the reflectiveness of senior staff in the ward, etc.) and other factors (e.g., time for the team to build a sense of safety with each other). This may also extend to other approaches of these groups, such as whether to have an external facilitator or not. We have therefore suggested that future studies could try to develop a theoretical model that explains how different factors interact to determine the best outcomes of these groups, which might be a phase-based or process-based model (lines 628-641).

---

## [Decision Letter · Decision Letter 1]

4 Dec 2024

Creating a Culture, Not Just a Space – a Qualitative Investigation into Reflective Practice Groups in Inpatient Mental Health Settings from the Perspectives of Facilitators and Attendees

PONE-D-24-15162R1

Dear Dr. Billings,

We’re pleased to inform you that your manuscript has been judged scientifically suitable for publication and will be formally accepted for publication once it meets all outstanding technical requirements.

Kind regards,

Alejandro Botero Carvajal, MD

Academic Editor

PLOS ONE

Reviewers' comments:

Reviewer's Responses to Questions

**Comments to the Author**

1. If the authors have adequately addressed your comments raised in a previous round of review and you feel that this manuscript is now acceptable for publication, you may indicate that here to bypass the “Comments to the Author” section, enter your conflict of interest statement in the “Confidential to Editor” section, and submit your "Accept" recommendation.

Reviewer #1: All comments have been addressed

2. Is the manuscript technically sound, and do the data support the conclusions?

Reviewer #1: Yes

3. Has the statistical analysis been performed appropriately and rigorously? 

Reviewer #1: N/A

4. Have the authors made all data underlying the findings in their manuscript fully available?

Reviewer #1: Yes

5. Is the manuscript presented in an intelligible fashion and written in standard English?

Reviewer #1: Yes

6. Review Comments to the Author

Reviewer #1: Thank you for addressing the comments thoroughly. I am happy to accept this article for publication.

7. PLOS authors have the option to publish the peer review history of their article (what does this mean?). If published, this will include your full peer review and any attached files.

Reviewer #1: **Yes: **Dr Sarah-Jane Fenton

---

## [Editor Report · Acceptance letter]

12 Dec 2024

PONE-D-24-15162R1 

PLOS ONE

Dear Dr. Billings, 

I'm pleased to inform you that your manuscript has been deemed suitable for publication in PLOS ONE. Congratulations! Your manuscript is now being handed over to our production team.

Kind regards, 

on behalf of

Dr. Alejandro Botero Carvajal 

Academic Editor

PLOS ONE